# Is Sleep Disruption a Cause or Consequence of Alzheimer’s Disease? Reviewing Its Possible Role as a Biomarker

**DOI:** 10.3390/ijms21031168

**Published:** 2020-02-10

**Authors:** Maria-Angeles Lloret, Ana Cervera-Ferri, Mariana Nepomuceno, Paloma Monllor, Daniel Esteve, Ana Lloret

**Affiliations:** 1Department of Clinical Neurophysiology, University Clinic Hospital of Valencia, Avda. Blasco Ibanez, 17, 46010 Valencia, Spain; malloretalc@gmail.com; 2Department of Human Anatomy and Embryology, Faculty of Medicine, University of Valencia, Avda. Blasco Ibanez, 15, 46010 Valencia, Spain; 3Department of Physiology, Faculty of Medicine, University of Valencia, Health Research Institute INCLIVA, Avda. Blasco Ibanez, 15, 46010 Valencia, Spain; mary.pinheiro@gmail.com (M.N.); paloma.monllor@uv.es (P.M.); daniel.esteve@ext.uv.es (D.E.); ana.lloret@uv.es (A.L.)

**Keywords:** REM, NREM, SWS, SWA, Spindles, CSF amyloid, CSF tau

## Abstract

In recent years, the idea that sleep is critical for cognitive processing has gained strength. Alzheimer’s disease (AD) is the most common form of dementia worldwide and presents a high prevalence of sleep disturbances. However, it is difficult to establish causal relations, since a vicious circle emerges between different aspects of the disease. Nowadays, we know that sleep is crucial to consolidate memory and to remove the excess of beta-amyloid and hyperphosphorilated tau accumulated in AD patients’ brains. In this review, we discuss how sleep disturbances often precede in years some pathological traits, as well as cognitive decline, in AD. We describe the relevance of sleep to memory consolidation, focusing on changes in sleep patterns in AD in contrast to normal aging. We also analyze whether sleep alterations could be useful biomarkers to predict the risk of developing AD and we compile some sleep-related proposed biomarkers. The relevance of the analysis of the sleep microstructure is highlighted to detect specific oscillatory patterns that could be useful as AD biomarkers.

## 1. Introduction

There is cumulating evidence that sleep quality and duration are relevant to cognitive processes. Cognitive impairments and sleep disturbances are often associated with several pathological conditions, although the nature of these relationships is still unclear and it is difficult to establish causal relations. Alzheimer’s disease (AD) is one of these conditions and it is also the most common form of dementia worldwide. Classically, it starts with a decline in episodic memory; however, neuropsychiatric symptoms can be present very early on and they can make an early diagnosis difficult [1,2].

AD presents a high prevalence and severity of sleep alterations [3,4,5] with research showing that sleep disturbances often precede in years the diagnosis of AD and that they might appear even before cognitive decline [6]. In fact, a longitudinal epidemiological study linked poor sleep quality in healthy people with cognitive impairment one year later [7]. Lim et al., in a community-based, prospective study, found that people with high sleep fragmentation had a 1.5-times higher risk of developing AD in a 6 years follow-up period [8]. Likewise, Benedict et al. conducted a 40-year study that evaluated 1574 men aged 50 and older, and observed that people with sleep disturbance had a 51% increase in the risk of developing AD [9]. Similarly, Bubu and colleagues estimated in a meta-analysis that the risk of dementia in patients with sleep disorders was 1.68 times greater [10]. Moreover, in a recent multi-center study, both midlife and late-life terminal insomnia were associated with a higher risk of dementia [11].

It is interesting to note that affected brain structures in people with disturbed sleep coincide with vulnerable areas in AD. Lower gray matter volume in hippocampus, precuneus, amygdala, and cingulate gyrus [12,13,14,15,16,17,18,19], and a higher degree of cortical atrophy have been described in cognitively-unimpaired insomnia patients [14,20]. A recent study performed in cognitively unimpaired adults aged between 45 and 75, found that insomnia patients presented decreases in grey matter volume in AD-related areas, that concur with other studies. Furthermore, they found greater volume in the left caudate in these subjects, which can also be seen in presymptomatic carriers of an AD genetic mutation [21]. Finally, a very recent study in middle-aged, cognitively unimpaired adults found lower grey matter volumes in the left angular gyrus, the bilateral superior frontal gyri, the thalami, and the right hippocampus in insomniac APOE-ε4 carriers (AD’s main risk factor) when compared to non-carriers with or without insomnia [21].

This temporal and anatomical relationship could suggest that sleep disturbances may influence or exacerbate AD pathology and that improving sleep may help slow down its progression [22,23]. Additionally, sleep alterations could be used to predict the risk of developing dementia and could also serve as early indicators of the disease. If sleep disorders constitute a risk factor to initiate or to accelerate the clinical course of AD, they could be used to develop disease biomarkers. Furthermore, a sleep biomarker with high sensitivity and specificity would be extremely useful, as electrophysiological recordings are noninvasive and relatively inexpensive.

In this manuscript we aim to review sleep disorders’ association with the neuropathological basis of AD, in an attempt to shed some light on the possible causal nature of this relationship and sleep’s plausible role as biomarker. We will first focus on the relevance of sleep for memory consolidation, followed by a brief review of sleep disturbances and of sleep and memory patterns with aging. Next, we will emphasize sleep disturbances in AD, either as clinical manifestations or as risk factors, and we will compile and discuss oscillatory, biochemical and system-based mechanisms linking sleep disturbances and the pathology of the disease. Finally, we will discuss the relationship between sleep disruption and gold standard AD biomarkers, as well as several sleep alterations that have been proposed as possible biomarkers for the early detection of AD.

## 2. Role of Sleep in Declarative Memory Consolidation

Declarative memory is hippocampal-dependent both in early stages of memory consolidation and also in recall [24] and after consolidation occurs, memories become hippocampal-independent [25]. Specifically in AD, declarative episodic memory is affected in the early stages, while other types of memory only deteriorate as the disease progresses [2,26,27,28].

The active systems consolidation hypothesis is currently one of the most important theories on the mechanisms of declarative memory consolidation. It focuses on the interaction between neocortex and hippocampus during sleep. After encoding, memories are labile and highly dependent on the hippocampus. To create more stable long-term memories, the hippocampal representations are repeatedly reactivated during sleep. These reactivations coactivate neocortical areas that will integrate the new representations into preexisting memories [29].

Ample evidence pointed out a key role of sleep in declarative memory consolidation. Animal and human studies supported three main evidences: firstly, the amount of sleep—or the amount of a specific sleep stage—increased following learning or environmental enrichment, with a consequent increase in hippocampal and cortical plasticity. Secondly, sleep—or certain sleep stages—improved performance in certain memory and learning tasks. Finally, sleep deprivation impaired cognitive processing in different ways and, in healthy humans, it decreased the ability to induce long term potentiation (LTP) plasticity [30,31].

Sleep in mammals consists of a cyclic alternation between Rapid Eye Movement (REM) and Non-REM (NREM) sleep. REM sleep was characterized by a theta activity (4–8 Hz) that appear especially in the hippocampus, coupled to a gamma activity (30–120 Hz). NREM sleep is divided into light (N1 and N2) and deep sleep (N3, also called slow wave sleep—SWS). During the N2 stage of NREM sleep, there is a characteristic thalamo-cortical reactivation that displays sudden bursts of oscillatory brain activity between 12–15 Hz called sleep spindles. N3 NREM sleep is characterized by slow waves between 1–4 Hz and by slow oscillations (SO) of <1 Hz, which represent the slow wave activity (SWA) [32]. During this stage an interplay between the hippocampus and neocortex occurs [33]. Finally, in N3 NREM also appears hippocampal sharp-wave/ripples, which are intermittent patterns of highly synchronous spiking seen as high-frequency oscillations (120–200 Hz) [34,35,36]. It represents the reactivation of hippocampal memory representations and have been proposed as a cognitive biomarker for episodic memory [37]. Sleep spindles also occur during N3 sleep where their occurrence is often obscured by slow oscillations [38] (see Figure 1 for summarizing).

The interaction and the timing of these oscillations during NREM sleep will be critical for a correct communication between the hippocampus, neocortex and other structures. A precise timing will allow thalamic spindles to couple with the correct SO phase. The specific SO-spindle coupling will then facilitate synaptic plasticity and enhance consolidation, while a mistimed coupling will diminish memory formation. At the same time, hippocampal sharp-waves/ripples need to be timed to spindles, which will result in a spindle-ripple coupling. Therefore, the correct timing and characteristics of these waves will allow the reactivated hippocampal information (ripples) to be transmitted to the neocortex (SO), by way of the thalamus (spindles). This process will facilitate synaptic consolidations processes that will store the information as a long-term memory [39].

Finally, REM sleep is thought to process specifically emotional memories, given that it presents a distribution of theta activity within limbic structures. Furthermore, REM sleep seems to be also critical for working and spatial memory consolidation [40], with a characterized coupling between theta and gamma oscillations. From this point of view, REM would mediate the integration and recombination of memory traces previously consolidated during NREM sleep [41].

## 3. Sleep and Memory in Sleep Disorders

Sleep is important for the consolidation of memory, therefore disturbed sleep might consequently result in its impairment. Here we will briefly discuss insomnia and obstructive sleep apnea syndrome (OSA), the two most frequent sleep disorders that are especially associated with AD. Insomnia and OSA have been shown to affect sleep-related memory consolidation, by reducing sleep time, altering sleep architecture, and/or producing fragmented sleep [43].

### 3.1. Insomnia

Insomnia is defined a difficulty initiating or maintaining sleep, or early morning awakening associated with an impairment during daytime [43]. The European prevalence varied between 5.7% to 19% [44] and its chronic form was related to a higher risk of cardiovascular diseases [45], depression [46], and cognitive impairment [47]. These risks were higher in patients that present short sleep duration, defined as less than 7 h of sleep per 24-h. Furthermore, not only insomnia frequently accompanies other disorders, such as multiple sclerosis, epilepsy, and AD, but it could also aggravate their symptoms [48].

In addition to the volume changes in brain areas related to AD that we have mentioned in the introduction, patients with insomnia showed a decline in activity in the hypothalamus, thalamus, ascending reticular activating system, insular cortex, amygdala, hippocampus, and in the anterior cingulate and medial prefrontal cortices. A finding that suggested an overactivity of the arousal and a dysregulation of the emotion and cognitive systems in insomnia [49].

Insomnia patients also presented reduced amounts of both SWS and REM sleep, shifts in sleep stages, fast electroencephalographic frequencies and an increased frequency of brief awakenings and microarousals [48]. These changes, especially the reduction of SWS, caused an impairment in sleep-dependent declarative memory consolidation [50].

### 3.2. Obstructive Sleep Apnea Syndrome

OSA is a chronic sleep disorder with a prevalence that varies from 9% to 38% in the overall population [51] and that increases with age [52]. OSA is characterized by repetitive episodes of partial (hypopnea) or complete (apnea) upper airway obstruction leading to intermittent hypoxia, hypercarbia, and arousals throughout the night [53]. Patients showed cortical thinning in areas such as the precentral motor gyrus, the postcentral sensory gyrus and insular and temporal cortices [54], which presents neuronal damage and atrophy, leading to cognitive decline.

OSA patients could present a significant reduction of sleep efficiency, defined as the percentage of time spent asleep while in bed. They also showed increased waking after sleep onset and arousal index, leading to sleep fragmentation [55]. Furthermore, these patients displayed a different time course of SWA, a decreased sleep spindle index in both SWS and N2 [56] and slow spindle frequencies [57].

These alterations caused cognitive impairments in many domains, such as attention, memory, executive function, psychomotor function and visuospatial function [53], which can lead to mild cognitive impairment (MCI) and dementia in the aging population [58].

## 4. Changes in Sleep and Memory in Normal Aging

With aging, several changes in the patterns and architecture of sleep occurred, which were especially perceptible after the sixth decade of life. In humans, normal aging was associated with advanced sleep phase, as older adults frequently present a phase-shift, falling asleep early in the night and waking up early in the morning [59,60]. Total nocturnal sleep time showed a near linear decrease of 8–12 min per decade [59,61,62,63]. Furthermore, sleep efficiency also displayed a slow but continuous decline after the sixth decade [62].

Regarding sleep stages, NREM-REM sleep cycles were fewer and shorter. Aging was characterized by a linear decrease in the proportion of REM [60,61] and a reduced amount of deep NREM sleep, compensated with an increased duration of light NREM [47,64,65,66]. These changes in sleep architecture during adulthood have been confirmed by two different meta-analyses, one of which included subjects from childhood to old age [61,62]. Light sleep predominance made the sleep more fragile, which leads to an increase in the number of arousals and nocturnal awakenings, as well as a longer duration of wake time after sleep onset [62,64]. Therefore, sleep maintenance was altered, resulting in less consolidated sleep and an increase in fragmentation. This reduction in nocturnal sleep time and quality was accompanied by increased daytime sleepiness and daytime naps [59].

As for sleep’s micro-architecture, SWA presented important reductions in amplitude, density and mean frequency in middle-aged adults, a deterioration that becomes especially prominent after 62 years old [65,67,68]. There was also a significant decline in N2 sleep spindle density with aging. Martin et al. (2013) found that age-related decline in density and amplitude of spindles was more prominent in anterior derivations, whereas duration decreased in the posterior derivations. Since each spindle characteristic seemed to present a distinct topographical pattern, they suggested that this topographical specificity could be a good biomarker to localize sleep’s age-sensitive changes [69].

These alterations had consequences on affective and cognitive processing, including memory and learning. Aging was related to impaired memory and there was ample evidence suggesting that age-related memory impairment is mediated by sleep disruption. The duration of SWS predicted the accuracy of memory after sleep in older adults, when a paradigm of word learning is used [70]. Likewise, the degree of SWA impairment predicted a worse consolidation of memory during nighttime sleep, resulting in a greater number of forgetfulness the following day [71,72]. These changes in sleep organization and their contribution to cognitive processing could partially explain the cognitive decline seen in the elderly.

Furthermore, the prevalence of sleep disorders, especially insomnia and sleep disordered breathing, increased with aging. [73,74,75]. These disturbances could lead to cognitive impairment as seen by cross-sectional studies, where older adults with insomnia performed significantly worse on memory span, integration of visual and semantic dimensions, and executive functioning task [76,77,78]. 

## 5. Sleep Disorders in Alzheimer’s Disease

Although sleep disturbances in AD patients were known for a long time, they used to be considered as just a consequence of the neurodegenerative process. However, current epidemiological studies showed that sleep disorders in AD patients go far beyond the physiological disturbances that occur in normal aging [10]. Early studies noted that sleep alterations were more prevalent amongst people with dementia than in the non-demented population [79]. This prevalence varied between 14–69% as was pointed out in a meta-analysis which included 5634 AD patients [2]. Another multicenter-retrospective study in Japan with 684 AD patients found the prevalence to be 21.3% [80]. Other studies found over 60% of patients with MCI and AD had at least one clinical sleep disorder [3,81] being insomnia and OSA the two most common. These discrepancies in prevalence could be related to the use of sleep questionnaires in many studies, as AD’s cognitive impairment might make it difficult for patients to reliably inform sleep disturbances [82].

Recent studies highlighted a higher prevalence and severity of sleep alterations in patients with an early-onset form of the disease [2,3]. Furthermore, sleep problems were generally associated with a worse evolution, including the development of more severe cognitive and neuropsychiatric symptoms, with a diminished quality of life and also with a higher caregiver burden [83,84,85]. A meta-analysis conducted by Bubu and colleagues showed that sleep alteration is associated not only with cognitive impairment or symptomatic AD, but also with changes in predictive biomarkers of people on the track to develop AD [10].

AD patients exhibited a significant alteration in the sleep/wake cycle, with an increase in the number of nighttime awakenings and a greater fragmentation of nocturnal sleep [79]. This nocturnal alteration was compensated with a much longer daytime sleep, consisting mainly of superficial sleep [86,87].

As for the sleep’s micro-architecture, nighttime sleep was also dominated by lighter sleep stages, with significant lesser percentages of SWS and REM sleep [87]. Importantly, when patients were separated by the disease severity, time spent on REM sleep tended to decrease as the disease progressed. Furthermore, a clear progression of the decrease in SWS was observed in parallel with cognitive decline [88,89]. We are going to analyze all these changes in depth in the next chapters.

## 6. Possible Mechanisms by Which Sleep May Participate in the Pathogenesis of AD

Dr. Alois Alzheimer described in 1907 the two major lesions found in the brains of AD patients, senile plaques and neurofibrillary tangles [90]. Senile plaques were mostly composed of beta-amyloid peptide (Aβ), that had a strong tendency to aggregate and precipitate, forming the typical plaques of the disease [91]. On the other hand, neurofibrillary tangles were formed mainly by a cytoskeleton protein called tau in a hyperphosphorylated state (P-tau). When tau was hyperphosphorylated the cytoskeleton was disrupted causing irreparable damage to neurons, which begin an apoptotic pathway that will lead to neurodegeneration [92].

### 6.1. Mechanisms Linking Sleep Respiratory Disorders with AD

According to the meta-analysis conducted by Bubu and colleagues (2017) mentioned before, OSA was the sleep problem most associated with a higher risk of AD [10]. It was associated with sleep fragmentation, daytime sleepiness and significant cardiovascular and metabolic disturbances. Furthermore, OSA might be a risk factor for developing MCI and dementia in the aging population, as it induced cognitive dysfunction in many domains, such as attention, episodic memory, working memory, and executive function [93].

Both AD and OSA were chronic diseases with a high prevalence and it was known that there was some overlap between both. In a meta-analysis, Emamian et al. found that patients with AD had a five-times greater risk of developing OSA than age-matched controls without cognitive impairment [94] and, in this line, it has been described that approximately 50% of the patients with AD will develop OSA after the diagnosis [95]. Conversely, patients with OSA obtained worse results in neuropsychological tests of executive functions, such as attention and memory. Importantly, patients with OSA carrying at least one APOEε4 allele, and therefore at risk of develop AD, presented lower scores than those with OSA that do not carry the gene [93]. In this regard, a prospective study with cognitively normal older women showed that those with severe sleep breathing disorder had an increased risk of developing MCI or dementia at follow-up [47]. Moreover, OSA was associated with an earlier age-of-onset of MCI and of progression to AD [96]. However, it seems that this risk had a gender difference, as a meta-analysis with almost twenty thousand subjects evidenced that sleep-related breathing disorders posed a significantly higher risk of cognitive decline for women than for man [97].

Recently, studies have seen an association between sleep-related breathing disorders and AD CSF biomarkers. While studying cognitively normal older adults, Osorio and colleagues (2014) found increases in CSF total-tau t(-tau), p-tau, and Aβ in OSA subjects without ApoE4 allele. Conversely, they showed that treating OSA could improve AD biomarkers, as these subjects showed an increase in SWS and a decrease in the levels of CSF Aβ when treated [98].

In addition to hypoxia, other factors such as progressive deterioration of sleep structure and sleep quality, as well as reduced brain flow, may contribute to cognitive impairment in patients with OSA and to an aggravation of AD in those patients with both pathologies [99].

### 6.2. Sleep and Amyloid Burden

So far, studies have shown a relationship between Aβ levels and sleep, however, whether it is causal or not is still controversial. A recent study evaluating sleep disorders using a murine model of AD, found that animals only presented sleep alterations after 6 months of age and that these alterations correlated to the performance on cognitive behavioral tests [100]. However, the authors did not measure Aβ cerebral levels and their conclusion that there were no sleep disorders until amyloid pathology appeared was based on an estimated age of pathology appearance. Conversely, another murine model study showed that alterations in both EEG and sleep architecture appeared before the deposit of Aβ in plaques, at 3–4 months of age [6].

Neurons do not release Aβ continuously but do so depending on their metabolic activity [101,102]. As CSF and blood metabolisms are generally regulated in a circadian manner, Aβ levels experiment cyclic fluctuations throughout the day [103,104]. Therefore, the concentration of this peptide in the brain’s extracellular space increases during active periods and decreases with rest [105,106]. This was reinforced by Kang et al., which, using a murine model, demonstrated that levels of Aβ in extracellular space correlated significantly with the time the animal was awake and that there was a negative correlation with sleep time, more accentuated for NREM sleep. In addition, sleep deprivation produced a clear increase of the previous day’s levels and increased the formation of Aβ plaques in the cerebral cortex [105].

In terms of human studies, sleep deprivation altered the levels of CSF Aβ in cognitively normal people, which could be seen after a single night of complete sleep deprivation [107,108]. Furthermore, sleep quality was worse in cognitively normal older adults with amyloid deposits than in those without it [109]. Moreover, reduced and fragmented SWS were associated with increases in CSF Aβ [110]. Of particular interest is a study by Mander at al. (2015) in cognitively normal older adults. In this study, the authors elaborated models to try to establish a causal relationship between Aβ accumulation in medial prefrontal cortex, reduction of SWA, and alteration of hippocampus-dependent memory consolidation [111]. They found that the best model reflected an indirect influence of Aβ deposits on the reduction of retentive memory, mediated by sleep. This result would imply a causal relation of Aβ deposits on sleep, and of sleep on memory disorder [111]. In AD Aβ accumulated in areas which generate NREM SWS [112], such as medial and lateral prefrontal cortex, posterior cingulate, and precuneus [113,114]. Interestingly, persons with higher cortical Aβ presence had equivalently worse hippocampus-dependent memory [111,115,116,117]. These findings could lead to the conclusion that NREM sleep is the link between Aβ accumulation and AD’s characteristic hippocampal-dependent memory loss.

Nevertheless, several authors pointed to a two-way relationship between sleep and amyloid pathology. Holtzman’s group postulated that sleep disturbance and increased wakefulness would lead to increased production and decreased clearance of Aβ. Furthermore, once Aβ is accumulated it results in more altered sleep, as evidenced in both mice and humans [118]. Meanwhile, Lucey et al. proposed that the initial appearance of Aβ would hinder the generation of slow oscillations in NREM sleep. This deficit would, in turn, increase the production of wakefulness-dependent Aβ and decrease sleep-dependent clearance, accelerating the deposit of Aβ and thus exacerbating the pathological cascade of AD [104,119].

### 6.3. Sleep and Tau Pathology

Very recent studies explored the relationship between tau pathology and different aspects of sleep. In the aforementioned work by Lucey et al. (2019), not only Aβ, but also tau pathology correlated to sleep disturbances in cognitively normal people and in those with MCI. The authors analyzed SWA in NREM sleep and showed an inverse relationship between AD pathology and SWA, especially in the lower range of the studied frequencies (1–2 Hz). Moreover, they found a stronger relationship between tau and SWA than between Aβ and SWA, leading them to conclude that tauopathy, rather than Aβ, correlates with sleep disturbances in asymptomatic or mildly symptomatic AD [119]. In another study by Kam et al. (2019), NREM sleep correlated with the amount of CSF Aβ, p-tau, and t-tau. Interestingly, of the three, t-tau was the one that most significantly associated with sleep spindle density, after adjusting for age, sex and ApoE4. Moreover, spindle duration, count, and density of fast spindles also correlated negatively with t-tau levels [120]. Lastly, Holth et al. (2019) found that the sleep-wake cycle and sleep deprivation influenced the amount of tau in interstitial fluid and CSF, both in experimental animals and in humans. Furthermore, they provided direct evidence that sleep disruption promoted the release and propagation of pathological tau aggregates in mice and that sleep deprivation in humans leads to an increase of more than 50% in CSF tau [121].

Tauopathy in AD seemed to begin very early in the locus coeruleus, where p-tau appeared decades before the first signs of cognitive impairment and, from there, it spread through axons to other memory related areas such as the entorhinal cortex and hippocampus [122]. With that in mind, Zhu et al. (2018) examined the effects of chronic sleep disruption on tauopathy in the locus coeruleus and hippocampus of transgenic mice that expressed a human tau protein with a mutation linked to AD. This study demonstrated that sleep disruption in early adulthood accelerated the deterioration of motor performance and increased neuronal loss in the locus coeruleus. Moreover, they found elevated levels of p-tau oligomers, which could still be observed 6 months after the sleep deprivation period, indicating that disturbances in this area were long maintained [123]. Other murine studies reinforce these results by evidencing that chronic sleep restriction may increase tau levels and that it may also impair hippocampus-dependent memory [124,125].

Lastly, it is recently proposed that an impairment in the SO-sleep spindle coupling might be a predictor of higher tau burden in the medial temporal lobe, but not of Aβ burden, while a diminished amplitude of <1 Hz SWA may be a predictor of Aβ burden [126].

In summary, there were evidences to support that sleep abnormalities increase both Aβ and tau and also that these pathological proteins may induce sleep disturbances. Therefore, it is important to emphasize the bidirectional role or positive feedback that occurs between these two processes.

### 6.4. Sleep and Glymphatic System

The transport of small metabolites and other molecules within the interstitial space determined the clearance of potentially neurotoxic peptides, such as Aβ and tau. CSF exchanged substances with the interstitial fluid, thus eliminating the products of cellular activity. It was likely that sleep, using this pathway, had the important function of removing multiple potentially toxic waste substances derived from neural metabolism, which may explain the restorative capacity of sleep [127].

Nevertheless, in the last decade, a new pathway for solute clearance has been proposed, in an attempt to better explain the relationship between sleep and the deposit of Aβ and other substances. Iliff’s group was the first to describe, in mice, a paravascular pathway that allowed the clearance of interstitial fluid through paravenous drainage [128]. In the central nervous system, vessels were surrounded by astrocyte podocytes that express the water channel aquaporin-4. Mice lacking AQP4 had reduced CSF influx through this paravascular system and a reduction of interstitial Aβ clearance. As a result of its analogy with the lymphatic system, this mechanism was called the “glymphatic system”.

The glymphatic activity seemed to be an important feature of the sleeping brain, rather than of the awake brain. Therefore, these changes in the paravascular flow could underlie the circadian fluctuations in extracellular and CSF Aβ levels observed in both rodents and humans [129]. In a study comparing awake mice with mice in states of physiological sleep or sleep induced by anesthesia, Xie and colleagues [130] found that Aβ was eliminated more efficiently (at twice the speed) during physiological sleep or anesthesia, than during wakefulness. Additionally, they evidenced that in the awake state there was a reduction in the cerebral interstitial space. These authors hypothesized that the contraction of the interstitial space in wakefulness increases the resistance of cerebral tissue to the flow of interstitial fluid and to entry of CSF. Furthermore, these alterations might have an adrenergic mechanism, as the administration of adrenergic antagonists in awake mice induced an increase of the affluence of tracers into the CSF and also of the volume of the interstitium, at levels comparable to those of sleep or anesthesia. Nevertheless, glymphatic activity seemed to decrease abruptly with aging, as evidenced by an 80–90% reduction of the glymphatic function in old mice [131], although more evidences and experiments are needed to extrapolate to humans.

Based on these results, many researchers are focusing their attention on this novel mechanism linking sleep, metabolite clearance and the development of AD [127,129,132,133,134], even though it has received contrary and critical opinions [135].

## 7. Sleep-Related Proposed Biomarkers in AD

Currently, evaluation of CSF Aβ and tau levels are AD’s gold standard biomarkers. However, this requires a lumbar puncture which is not only an unpleasant procedure, but it is also invasive. For this reason, it is important to investigate new reliable and noninvasive biomarkers, and some specific sleep abnormalities could be good candidates.

As mentioned earlier, sleep stages are characterized by specific oscillatory activities which are relevant to memory processing, neuronal rest and maintenance of homeostasis. Thus, in the search of useful new biomarkers, it is interesting to analyze the changes that appear in certain oscillatory patterns and their relationship with gold standard biomarkers.

### 7.1. General EEG Features

Early studies in AD patients demonstrated a slowing of the awake and REM sleep occipital EEG (from 9.40 Hz to 6.40 Hz), the occurrence of paroxysmal sharp waves in quiet wakefulness or during sleep and “poorly formed or absent” spindle activity [88]. In general, the reduction of deep sleep, the increase of awakenings and the progressive slowing of EEG activity were seen in normal aging but were remarkably accentuated in AD patients.

### 7.2. SWA in NREM

Studies in healthy elderly adults found that both diminished SWA and SWA disruption were associated with elevated CSF Aβ [99,110,136]. Likewise, Mander and colleagues (2015) observed that NREM sleep SO correlated with the deposits of Aβ. Given that the association between amyloid pathology and NREM sleep disturbances occurred not only in MCI and AD, but also in older subjects in the preclinical phase, the authors proposed that a reduction of < 1 Hz slow oscillatory activity index in NREM sleep may constitute a new biomarker of preclinical AD [111]. Specifically, diminished N3 slow wave oscillations were associated with elevated CSF Aβ42 [99,110], and slow wave activity disruption during polysomnogram induced increases of CSF Aβ 40 [136]. The aforementioned work by Osorio and colleagues (2014) provided useful objective measures in the search for biomarkers by combining polysomnography and CSF measures, studying cognitively normal elderly subjects [98]. They found a significant inverse correlation between CSF Aβ42 levels, SWS duration and other SWS parameters. In particular, total SWA in the frontal electrodes was the best predictor of reduced CSF Aβ42 levels when controlling for age and ApoE status. Remarkably, they did not find correlation between CSF Aβ42 with total sleep time or time in N1, N2, or REM sleep. Therefore, reduced and fragmented SWS correlated with higher levels of CSF Aβ_42_. On the light of their findings, the authors suggest that disturbed deep sleep might have a causal relationship to the pathogenesis because of an increase in soluble Aβ levels prior to amyloid deposition [110].

On the other hand, Lucey and colleagues found a stronger inverse relationship between SWA, especially for frequencies of 1–2 Hz, and tau rather than Aβ in asymptomatic or mildly symptomatic AD. So, the authors proposed that SWA reduction in NREM may be an early biomarker of AD and that it could also serve to monitor the progression of AD pathology and the response to treatment, given that its measurement does not require invasive techniques [104].

### 7.3. Other Possible Biomarkers

A very recent research, performed on cognitively normal elderly individuals, found a significant negative correlation between spindle density in N2 sleep and CSF t-tau, p-tau, and Aβ. This correlation was especially strong with t-tau as well as with the ratio of t-tau/Aβ. Nevertheless, CSF tau measures did not correlate with total sleep time nor with sleep quality [120]. So, spindle density in N2 could be another candidate as AD biomarker.

On the other hand, while SO measurements correlated very well with Aβ, it was the SO-Spindle coupling that predicted the greatest accumulation of tau specifically in the medial temporal lobe [126]. Considering that coupling of SO-spindles was involved in sleep-dependent hippocampal memory processing [137], was not surprising its contribution to AD-related cognitive decline.

In studies with transgenic mice carrying human amyloid precursor protein (h-APP), although no differences were found in net theta power nor gamma power, theta-gamma coupling was found to be dramatically reduced [138]. Other studies have evidenced further gamma coupling alterations [139]. However, to date, no studies have described similar data in human REM sleep, when theta and theta-gamma coupling were predominant [140]. It would be worthwhile to evaluate these issues in humans, seen as REM sleep disruption forecasted worse Mini-Mental State Examination scores in MCI and AD patients [141,142,143].

Finally, molecules implicated in the glymphatic system could be another line of research of future sleep-related biomarkers in AD pathology (Table 1).

## 8. Could Sleep Oscillatory Activities be Good Biomarkers for AD?

The criteria for establishing a good biomarker for the diagnosis of AD include: to reflect a pathophysiological process of the disease; to display high sensitivity and specificity for AD; to react upon pharmacological intervention; and to be reproducible, noninvasive, inexpensive, and rapid [144]. Therefore, specific sleep alterations might be good candidates since they reflect the pathological brain processes related to AD, that is, they correlate to Aβ and tau levels. It is important to highlight that a good biomarker should correlate with the two main pathological signatures of AD independently, Aβ and p-tau, and that is the case with these sleep alterations. Furthermore, their evaluation is reproducible, noninvasive, and cost-effective, as the development of portable machines makes polysomnography a non-complicated technique that can be performed wherever the patient is.

Nevertheless, their sensitivity and specificity for AD has yet to be analyzed and also their changes related to a pharmacological intervention. The most recent studies point to the higher utility of studying specific oscillatory patterns rather than general data or even visual analysis of the sleep microarchitecture. More studies are needed to evaluate the response of sleep-related possible biomarkers to such an intervention, as current medications used to aid sleep are not designed to improve these specific parameters.

Moreover, many neurologic and psychiatric diseases present sleep disorders. Therefore, in order to reach higher specificity, it is key to detect changes in the microarchitecture of sleep oscillations that only occur in AD. Such details could be better controlled in animal models of AD and further validated in human studies. In addition, given AD presents higher prevalence in women, the design of sleep studies should consider gender differences in sleep parameters.

Should these obstacles be overcome, sleep-related biomarkers could have a promising role as a tool to support the diagnosis of AD in patients with cognitive decline, as well as to evaluate disease progression, as they seem to correlate well with disease severity and cognitive decline.

## 9. Conclusions

Sleep alterations occur both in normal aging and in AD and they include a decrease in total sleep time, REM sleep, and deep NREM sleep (SWS), and an increase in sleep fragmentation and in the time spent in light NREM phases. However, some specific oscillatory patterns (reduction of SWA and of spindle density and impairment in the slow oscillation-sleep spindle coupling as well as theta-gamma coupling) are associated with an increase in biomarkers of AD neuropathology. Evidence supports a bidirectional relationship between sleep abnormalities and Aβ and tau burden, meaning that sleep disruption might be both a cause and a consequence of AD. Nevertheless, much needs still to be learned about these links, especially after the glymphatic system has been described as a possible new pathway linking sleep and AD. Lastly, specific oscillatory patterns linked to sleep disruption seem to be good candidates as biomarkers to help in the early diagnosis of the disease and to track its evolution, although more studies are still needed before considering them as useful biomarkers in clinical practice. See Figure 2 for a schematic rendition summarizing the main features relating AD and sleep disorders.

## Figures and Tables

**Figure 1 ijms-21-01168-f001:**
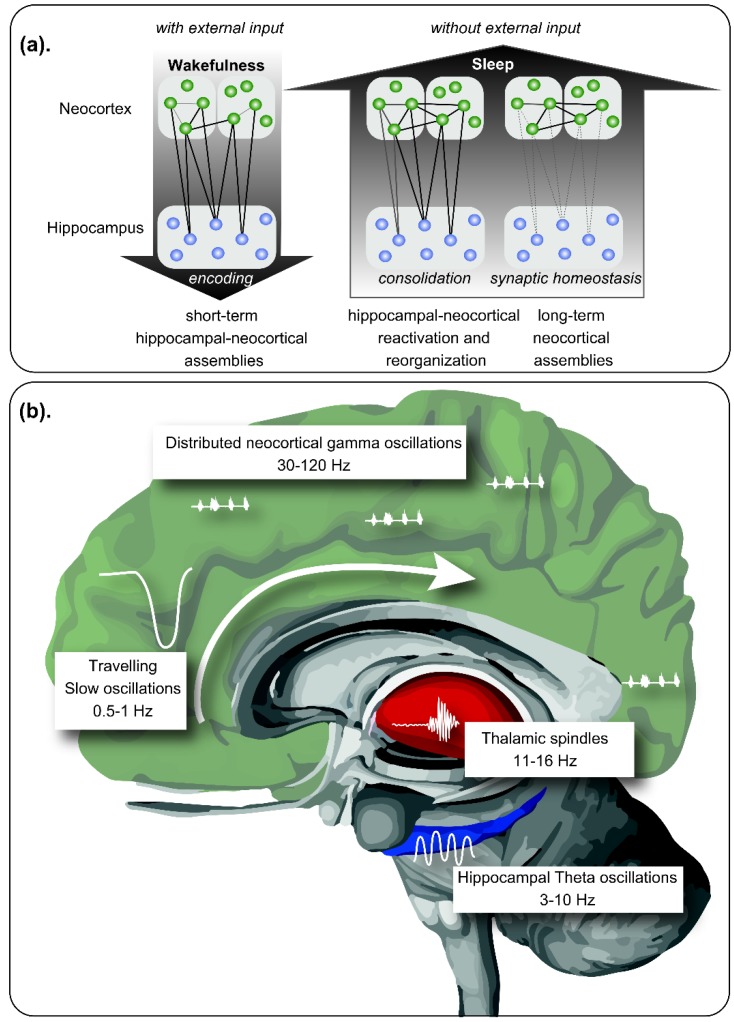
Role of sleep in memory consolidation. (**a**) During wakefulness, new memories are encoded by neuronal assemblies distributed within the neocortex and the hippocampal formation, integrating contextual information. During sleep, in absence of external inputs, memory reorganization is performed, reinforcing some neuronal assemblies by reactivation of the hippocampal-neocortical and thalamocortical circuits. Neural plasticity allows the consolidation of the stronger hippocampal-independent neocortical connections, together with the extinction of weak memory traces. (**b**) Neural oscillations hierarchically coupled allow memory processing during sleep. Slow oscillations propagate across cortical areas during Non-Rapid Eye Movement (NREM) sleep following a craneo-caudal direction, exhibiting upstates and downstates. During the downstates, neuronal rest is observed, while during upstates, highly synchronous discharges are observed, with replay of previous neuronal assemblies, coupled with sleep spindles originated in the thalamus. REM sleep is characterized by prominent theta waves in the hippocampal formation among other structures, coupled to gamma activity distributed within the neocortex. In this period, place-cells and grid cells exhibit a replay, allowing consolidation of contextual memory. Modified from Born and Wilhelm (2012) [42].

**Figure 2 ijms-21-01168-f002:**
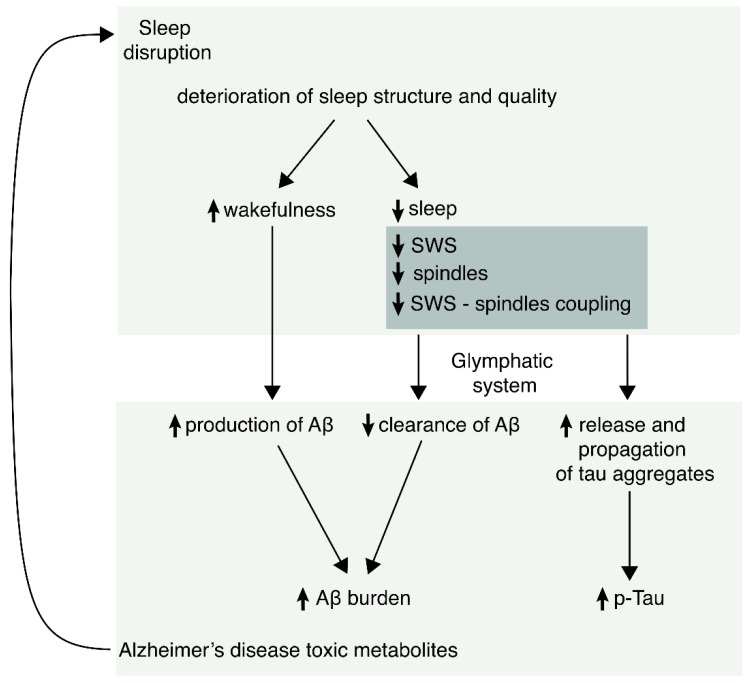
Schematic rendition summarizing the possible sleep biomarkers in AD. Experimental evidence suggests that sleep disorders appear early in AD. Sleep abnormalities induce more formation of pathological tau and beta-amyloid peptide (Aβ). Likewise, both Aβ and P-tau induce sleep disorders, generating a positive feedback process.

**Table 1 ijms-21-01168-t001:** Current evidence of possible sleep biomarkers and their association with gold standard Alzheimer’s disease (AD) biomarkers.

Oscillatory Parameter	Aβ Burden	Tau Pathology	AD-Related Localization
NREM SWA (< 1 Hz)	✓		Prefrontal Cortex Posterior cingulate Precuneus CSF
NREM SWA (1–4 Hz)		✓	Entorhinal Parahippocampal Orbital frontal Precuneus Inferior parietal Inferior temporal
Spindles	✓	✓	CSF
SO-Spindle coupling		✓	Medial temporal lobe

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
