# Peer review of "Is Sleep Disruption a Cause or Consequence of Alzheimer’s Disease? Reviewing Its Possible Role as a Biomarker"

_ijms, 2020, doi:10.3390/ijms21031168_

Round 1

Reviewer 1 Report

This is an interesting review at a time when there is much interest in the sleep and AD. It would benefit with some improvement in the language generally and simplify sentences.

I have few comments:

1. Title: The title needs to say its literature review.

2. Abstract:  The language needs to be simplified. The 3rd sentence rephrased for clarity.

Main document:

3. Please expand SIESTA on page 4, line 165 and J-BIRD, line 218.

4. To reduce sections 2 and 3 as they are very elaborate and also 1st paragraph of section 4 needs to be concise.

5. ‘Sleep disorders in AD’ could be a separate section instead of being under section 3.

6. It would be helpful to have a table with the current evidence of sleep biomarkers, and their association with other markers (CSF, imaging, behaviour tests), would be helpful to the readers.

Author Response

This is an interesting review at a time when there is much interest in the sleep and AD. It would benefit with some improvement in the language generally and simplify sentences.

I have few comments:

Title: The title needs to say its literature review.

We thank the reviewer his/her kind comments and suggestion. We have changed the title.

Abstract:  The language needs to be simplified. The 3rdsentence rephrased for clarity.

We have substantially changed the Abstract Section. We hope now is clearer than before.

Main document:

Please expand SIESTA on page 4, line 165 and J-BIRD, line 218.

Since general comments indicated that our manuscript need to be clearer, in order to clarify information and to be concise we have decided to delete the reference to these studies.

To reduce sections 2 and 3 as they are very elaborate and also 1stparagraph of section 4 needs to be concise.

We thank the reviewer his/her suggestion. We have rewritten chapters 2 and 3 summarizing information. However, following the suggestion of the reviewer 2 we have added some more information to these chapters and we have included a new one.

‘Sleep disorders in AD’ could be a separate section instead of being under section 3.

We apologize for the mistake and we have corrected this error. Sleep disorder in AD is now a new chapter.

It would be helpful to have a table with the current evidence of sleep biomarkers, and their association with other markers (CSF, imaging, behaviour tests), would be helpful to the readers.

We thank the suggestion. We have added a table with AD markers (imaging, Ab and tau) related to sleep oscillations alteration.

Reviewer 2 Report

In general the paper is difficult to follow due to deficits in the flow of information and lack of clarity on the relevance of some of the information to the subject of the matter. The authors are advised to rearrange the paper so that the flow will be kept from normal function of sleep in memory consolidation (including brain areas and sleep states involved) into the relationship between sleep and memory in sleep disorders, into the relationship between sleep and memory in normal aging and into the relationship between sleep and memory in MCI and AD.

General comment: There is no reference to the brain areas affected by dementia vs those affected by insomnia. Also it is unclear what type of memory is mostly lost in AD and how does this connect to the pathological and physiological changes in brain structures.

In all cases effort should be made to provide epidemiological data on sleep vs memory impairments so that the reader will have a better understanding on the co-occurrence of these impairments and their relevance to the progression of dementia.

The attempt to define markers is fine but also consider including information on how these markers could be related to the specific changes in brain structure in dementia vs insomnia.

Figure 1 has inconsistencies and unhelpful. Consider major revision (see for example review by Björn Rasch and Jan Born About Sleep's Role in Memory Physiol Rev. 2013 Apr; 93(2): 681–766.)

Figure 2 is not comprehensive and inconsistent, e.g. no description of non-declarative/(procedural ) memory missing

Figure 3 is not needed. The text is elaborate enough

Figure 4 is fine.

The references in the list seem to be incomplete in many cases

Specific comments:

The introduction may be improved by relating to more recent reviews such as Jens G. Klinzing, Niels Niethard & Jan Born  Mechanisms of systems memory consolidation during sleep Nature Neuroscience volume 22, pages1598–1610(2019)

line 39 To show comorbidity or causal relationship it will be useful to include information on  % AD patients having insomnia/short sleep vs age-matched non-AD population and % insomnia patients who develop AD in say 10 years compared to age matched non-insomnia subjects. Also it is important to explain what is considered short sleep duration is.

Line 46 to be useful as marker we need to have information on sensitivity and specificity

Line 51: sleep and memory patterns with aging

Line 65-74 unclear please rephrase.

Line 79 Much of the information is about retention of memory traces post learning after sleep vs sleep deprivation

Line 85 jumping to AD amidst discussion of normal memory mechanisms

Line 93-96 it is unclear whether you are relating to different phases of memory consolidation or different fate of such memories?

Line 11-117 It would be good to align the Introduction with the specific processes that are affected by sleep in memory consolidation (type of memory, brain regions involved, route of consolidation, timing etc).

Line 124-126 in part out of context here.  

Line 127 what phase?

Line 146-155 should look more specifically at good-sleepers vs insomnia patients. The most consistent change with age in good sleepers is in %S3 (line 172-175).  Attention should be given to comorbidities, effects of medications which also affect sleep but are not part of healthy aging

Lines 176-178 The Up and Down states of SWS are not very well defined.

Lines 188-195 More information on insomnia and memory in cognitively intact people is missing

Lines 197-202 more epidemiological data is useful here

Line 308-316 What are the brain areas involved in amyloid accumulation? Tauopathy? How are they related to sleep? A schematic presentation will be helpful

Line 368-369 This is hypothesis generating. Need to be careful with extrapolations to humans because mice do not have AD.

Author Response

In general, the paper is difficult to follow due to deficits in the flow of information and lack of clarity on the relevance of some of the information to the subject of the matter. The authors are advised to rearrange the paper so that the flow will be kept from normal function of sleep in memory consolidation (including brain areas and sleep states involved) into the relationship between sleep and memory in sleep disorders, into the relationship between sleep and memory in normal aging and into the relationship between sleep and memory in MCI and AD.

We thank the reviewer for his/her deeply revision of our manuscript. We have substantially modified it. We have deleted some information to clarify concepts, re-located paragraphs and added some more data. We have also added a new section about memory in sleep disorders and a summarizing table at the end. We hope the information flows now and the review is clearer and improved.

General comment: There is no reference to the brain areas affected by dementia vs those affected by insomnia. Also, it is unclear what type of memory is mostly lost in AD and how does this connect to the pathological and physiological changes in brain structures.

We have added information about brain areas affected in insomnia and, at the same time, vulnerable for AD.

In all cases effort should be made to provide epidemiological data on sleep vs memory impairments so that the reader will have a better understanding on the co-occurrence of these impairments and their relevance to the progression of dementia.

We thank the reviewer for the suggestion. We have added a new chapter entitled: “Sleep and memory in sleep disorders”.

The attempt to define markers is fine but also consider including information on how these markers could be related to the specific changes in brain structure in dementia vs insomnia.

We have added information throughout the text about AD vulnerable areas (presence of Ab or tau) and sleep disorders. We have also added a table at the end, with AD markers (imaging, Ab and tau) related to sleep oscillations alteration. It was also suggested by reviewer 1.

Figure 1 has inconsistencies and unhelpful. Consider major revision (see for example review by Björn Rasch and Jan Born About Sleep's Role in Memory Physiol Rev. 2013 Apr; 93(2): 681–766.)

We thank the reviewer for his/her exhaustive revision. We have changed figure 1 based on the figure recommended by the reviewer. We hope now is more informative and helpful.

Figure 2 is not comprehensive and inconsistent, e.g. no description of non-declarative/(procedural) memory missing

We have deleted figure 2.

Figure 3 is not needed. The text is elaborate enough

We have also deleted figure 3.

Figure 4 is fine.

The references in the list seem to be incomplete in many cases

We apologize for the mistake and we have checked the reference list.

Specific comments:

The introduction may be improved by relating to more recent reviews such as Jens G. Klinzing, Niels Niethard & Jan Born Mechanisms of systems memory consolidation during sleep Nature Neuroscience volume 22, pages1598–1610(2019)

We have added more recent reviews as the suggested, in the chapter “Role of sleep in declarative memory consolidation”.

line 39 To show comorbidity or causal relationship it will be useful to include information on % AD patients having insomnia/short sleep vs age-matched non-AD population and % insomnia patients who develop AD in say 10 years compared to age matched non-insomnia subjects. Also it is important to explain what is considered short sleep duration is.

We have added some more studies and data in the Introduction as suggested.

Line 46 to be useful as marker we need to have information on sensitivity and specificity

We thank the reviewer for this important point. We have discussed it in depth in the “Biomarker Section” at the end of the review.

Line 51: sleep and memory patterns with aging

Thank you for the suggestion. We have changed the sentence.

Line 65-74 unclear please rephrase.

Thank you for the comment. We have deleted this part following the recommendation of reviewer 1 about summarizing information.

Line 79 Much of the information is about retention of memory traces post learning after sleep vs sleep deprivation

We agree with the reviewer that these sentences could be clearer. However, following the recommendation of reviewer 1 we have deleted it in order to be concise.

Line 85 jumping to AD amidst discussion of normal memory mechanisms

We agree with the reviewer and we have re-located the sentence to a more appropriate section. 

Line 93-96 it is unclear whether you are relating to different phases of memory consolidation or different fate of such memories?

We agree with the reviewer that these sentences could be improve but we have deleted it to clarify and concise the text as indicating reviewer 1.

Line 11-117 It would be good to align the Introduction with the specific processes that are affected by sleep in memory consolidation (type of memory, brain regions involved, route of consolidation, timing etc).

We have added the information required.

Line 124-126 in part out of context here.  

We agree with the reviewer and we have deleted it.

Line 127 what phase?

We have changed the sentence.

Line 146-155 should look more specifically at good-sleepers vs insomnia patients. The most consistent change with age in good sleepers is in %N3 (line 172-175).  Attention should be given to comorbidities, effects of medications which also affect sleep but are not part of healthy aging

We have added information about insomnia and aging.

Lines 176-178 The Up and Down states of SWS are not very well defined.

We thank the reviewer for his/her comment. In order to clarify concepts and to be more concise, we have deleted Up and Down states information.

Lines 188-195 More information on insomnia and memory in cognitively intact people is missing

We have added a new chapter “Sleep and memory in sleep disorders” including information about insomnia and obstructive sleep apnea syndrome patients, the two most frequent sleep disorders that are especially associated with AD.

Lines 197-202 more epidemiological data is useful here

We have added more references and epidemiological data about insomnia and AD in the “Introduction” and in the “Sleep Disorders in Alzheimer's Disease” section.

Line 308-316 What are the brain areas involved in amyloid accumulation? Tauopathy? How are they related to sleep? A schematic presentation will be helpful

We have added information about brain areas with accumulation of A or tau related to sleep. We also have added a table with AD markers (imaging, Ab and tau) associated with sleep oscillations alteration.

Line 368-369 This is hypothesis generating. Need to be careful with extrapolations to humans because mice do not have AD.

We agree with the reviewer and we have clarified this concept.

Round 2

Reviewer 2 Report

The paper has improved significantly.

One minor comment is the Abstract 

Lines 16-17 ……includes both   cognitive impairments and sleep disturbances with high comorbidity. Cognitive impairments are the main AD symptoms but insomnia is not included in AD symptoms. These are  comorbidities. Please repharse this sentence

Author Response

Comments and Suggestions for Authors:

The paper has improved significantly.

One minor comment is the Abstract

Lines 16-17 ……includes both cognitive impairments and sleep disturbances with high comorbidity. Cognitive impairments are the main AD symptoms but insomnia is not included in AD symptoms. These are comorbidities. Please repharse this sentence

We appreciate the contribution of the reviewer to the improvement of our manuscript.

We have corrected the abstract according to the comments. The text has been changed to:

"Alzheimer’s disease (AD) is the most common form of dementia worldwide and presents a high prevalence of sleep disturbances. "